# Interpreting Learned Feedback Patterns in Large Language Models

**Luke Marks**[*†]   **Amir Abdullah** [*†◇]   **Clement Neo**[†]   **Rauno Arike**[†]

**David Krueger**[⊙]   **Philip Torr**[‡]   **Fazl Barez**[*†‡]

[†]Apart Research   [◇]Cynch.ai   [⊙] University of Cambridge

[‡]Department of Engineering Sciences, University of Oxford

## Abstract

Reinforcement learning from human feedback (RLHF) is widely used to train large language models (LLMs). However, it is unclear whether LLMs accurately learn the underlying preferences in human feedback data. We coin the term *Learned Feedback Pattern* (LFP) for patterns in an LLM's activations learned during RLHF that improve its performance on the fine-tuning task. We hypothesize that LLMs with LFPs accurately aligned to the fine-tuning feedback exhibit consistent activation patterns for outputs that would have received similar feedback during RLHF. To test this, we train probes to estimate the feedback signal implicit in the activations of a fine-tuned LLM. We then compare these estimates to the true feedback, measuring how accurate the LFPs are to the fine-tuning feedback. Our probes are trained on a condensed, sparse and interpretable representation of LLM activations, making it easier to correlate features of the input with our probe's predictions. We validate our probes by comparing the neural features they correlate with positive feedback inputs against the features GPT-4 describes and classifies as related to LFPs. Understanding LFPs can help minimize discrepancies between LLM behavior and training objectives, which is essential for the safety and alignment of LLMs.

## 1   Introduction

Large language models (LLMs) are often fine-tuned using reinforcement learning from human feedback (RLHF), but it is not understood whether RLHF results in LLMs accurately learning the preferences that underlie human feedback data. We refer to patterns in an LLM's activations learned during RLHF that enable it to perform well on the task it was fine-tuned for as the LLM's *Learned Feedback Patterns* (LFPs). Formally, for an input $\mathbf{X}$ and activations $\mathbf{H}(\mathbf{X}, \theta)$ from a fine-tuned LLM parameterized by $\theta$, we describe its LFPs as the differences in $\mathbf{H}(\mathbf{X}, \theta)$ caused by training $\theta$, that result in the outputs performing better under the fine-tuning loss. LFPs are a major component of what an LLM has learned about the fine-tuning feedback.

For example, consider a sentiment analysis task where the ground truth dataset labels the word "precious" as having positive sentiment. However, the fine-tuned LLM's activations, when probed, predict negative sentiment. This discrepancy, where the LLM's output would receive negative feedback according to the true preferences, is an example of divergence between LFPs and the preferences underlying the human feedback data used in fine-tuning.

---

[*] Equal Contribution

38th Conference on Neural Information Processing Systems (NeurIPS 2024).

Our objective is to study and measure this divergence. However, obstacles like feature superposition [12] in dense, high dimensional activation spaces, and limited model interpretability obscure the relationship between human-interpretable features and model outputs. In this paper we ask: **Can we measure and interpret the divergences between LFPs and human preferences?**

Continued deployment of LLMs fine-tuned using RLHF with greater capabilities could amplify the impact of LFPs divergent from the preferences that underlie human feedback data. Possible risks include manipulation of user preferences [1] and catastrophic outcomes when models approach human capabilities [10]. The ability to measure and explain the divergences of LFPs in human-interpretable ways could help minimize those risks and inform developers of when intervention is necessary. To achieve this, we extend existing research that uses probes to uncover characteristics of larger, deep neural networks [2, 5, 27]. Our probes are trained on condensed representations of LLM activations. The trained probes predict the feedback implicit in condensed LLM activations. We validate our probes by comparing the features they identify as active in activations with implicit positive feedback signals against the features `GPT-4` describes and classifies as being related to the LFPs.

The decoders of autoencoders trained on LLM activations with a sparsity constraint on the hidden layer activations have been shown to be more interpretable than the raw LLM weights, partially mitigating feature superposition [29, 9, 11]. The outputs of these autoencoders comprise the condensed representations of LLM activations. By training our probes on sparse autoencoder outputs, we make it easier to understand which features in the activation space correlate with implicit feedback signals.

We hypothesize that consistent patterns in the activations of fine-tuned LLMs correlate with the fine-tuning feedback, allowing the prediction of the feedback signal implicit in these activations. In validation of this hypothesis, we make the following contributions:

- We use synthetic datasets to elicit activation patterns in fine-tuned LLMs related to their LFPs. We make these datasets publicly available for reproducibility and further research.
- We train probes to estimate the feedback signal implicit in a fine-tuned LLMs activations (§3.3).
- We quantify the accuracy of the LFPs to the fine-tuning feedback by contrasting the probe's predictions and the true feedback (§3.3).
- We use `GPT-4` to identify features in the fine-tuned LLM's activation space relevant to the LFPs. We validate our probes against these feature descriptions, showing that the two methods attribute similar features to the generation of outputs that receive a positive feedback signal. (§3.4).

Code for all of our experiments is available at https://github.com/apartresearch/Interpreting-Learned-Feedback-Patterns.

## 2 Background and related work

We study LLMs based on the Transformer architecture [37]. Transformers operate on a sequence of input tokens represented by the matrix $\mathbf{X} \in \mathbb{R}^{L \times d}$, where $L$ is the sequence length and $d$ is the token dimension. For each token a query $\mathbf{Q}$, key $\mathbf{K}$ and value $\mathbf{V}$ is formed using the parameter matrices $\mathbf{W}_q, \mathbf{W}_k, \mathbf{W}_v \in \mathbb{R}^{d \times d}$, giving $\mathbf{Q} = \mathbf{W}_q \mathbf{X}$, $\mathbf{K} = \mathbf{W}_k \mathbf{X}$, and $\mathbf{V} = \mathbf{W}_v \mathbf{X}$. The attention scores, $\mathbf{A} = \text{softmax}\left(\frac{\mathbf{Q}\mathbf{K}^\top}{\sqrt{d}}\right)$, measure the relevance of each token to every other token. The final output is obtained by weighting the values by the attention scores, resulting in the output matrix $\mathbf{O} = \mathbf{A}\mathbf{V}$. $\mathbf{O}$ is then passed through a multi-layer perceptron (MLP) and combined with the original input via a residual connection, forming the final output of the layer and the input for the next layer.

There is a significant body of evidence that deep neural networks such as Transformers learn human-interpretable features of the input, providing a strong motivation for interpretability research [26, 28, 21, 7]. However, there is often not a one-to-one correspondence of neurons and features. When multiple features in a single neuron are represented near-orthogonally, this phenomenon is known as 'superposition' [12], allowing models to represent more features than dimensions in their activation space. This can be practical when those features are sparsely present in training data. Superposition poses a major obstacle to neural network interpretability, and this is expected to extend to the study of LFPs learned during RLHF. A promising approach to disentangling superposed

features in neural networks is to train autoencoders on neuron activations from those networks. Given encoder weights $\mathbf{W}_E \in \mathbb{R}^{n \times h}$, decoder weights $\mathbf{W}_D \in \mathbb{R}^{h \times n}$, a bias vector $b_{\mathbf{E}} \in \mathbb{R}^h$, and an input $\mathbf{X} \in \mathbb{R}^{m \times n}$, the output $\hat{\mathbf{X}}$ is computed as:

$$\hat{\mathbf{X}} = \mathbf{W}_D(\text{ReLU}(\mathbf{W}_E\mathbf{X} + b_{\mathbf{E}})) \tag{1}$$

The sparse autoencoder loss function is typically:

$$L(\mathbf{X}) = \frac{1}{|\mathbf{X}|} \sum_{\mathbf{X} \in \mathbf{X}} \left\| \mathbf{X} - \hat{\mathbf{X}} \right\|_2^2 + \alpha \left\| \text{ReLU}(\mathbf{W}_E\mathbf{X} + b_{\mathbf{E}}) \right\|_1 \tag{2}$$

Where the first term penalizes the Euclidean distance between $\hat{\mathbf{X}}$ and $\mathbf{X}$, and the $\ell_1$ term encourages the output $\hat{\mathbf{X}}$ to be a sparse linear combination of features in $\mathbf{W}_D$, providing an interpretable overview of the superposed features that were active in the autoencoder's input. $\hat{\mathbf{X}}$ is then a condensed representation of $\mathbf{X}$.

Early results suggest sparse autoencoders can recover features of the input even when those features are represented in a superposed manner [9, 11]. Sharkey et al. [34] train two sparse autoencoders with different hidden sizes, and find similar features in both. Because the features learned by an autoencoder are sensitive to its hidden size, finding similar features in both autoencoders increases the likelihood that they are actually features of the input. Other works exploring related techniques include Yun et al. [41], who apply sparse dictionary learning to visualize the residual streams of Transformer models, and Gurnee et al. [16], who find human-interpretable features in LLMs using sparse linear probes.

Even when features are interpretable by humans, it can be laborious for a human labeller to identify plausible descriptions of what a neuron represents. Recent work has shown that this can be automated at scale [13, 7]. Bills et al. [7] provide `GPT-4` with a set of activations discretized and normalized to a range of 0 and 10 for a set of tokens passed to the model as a prompt. `GPT-4` then predicts an explanation for what the neuron represents based on those activations, and predicts discretized activations for tokens as if that description were true. The efficacy of a neuron explanation is judged by the Pearson correlation coefficient of the predicted and true activations.

To our knowledge, no general methods have been proposed for finding human-interpretable representations of LFPs learned via RLHF. Previous literature on reward model interpretability has focused on more conventional RL methods. For example, Jenner and Gleave [20] provide a framework for preprocessing reward functions learned by RL agents into simpler but equivalent reward functions, which makes visualizations of these functions more human-understandable. Michaud et al. [25] explain the reward functions learned by Gridworld and Atari agents using saliency maps and counterfactual examples, and find that learned reward functions tend to implement surprising algorithms relying on contingent aspects of the environment. Gleave et al. [14] and Wolf et al. [40] present methods for comparing and evaluating reward functions learned through RL training without requiring these functions to be human-interpretable. Probing deep neural networks using linear classifiers is well-established [2, 5, 27], but the architecture of prior probes varies considerably to our approach, mostly in the objective of the probe. We specifically analyze the implicit representation of feedback in activations over contrastive inputs.

## 3 Experiments and methodology

Here, we detail each stage of our experimental pipeline. The major steps are as follows:

1. Fine-tune pre-trained LLMs using RLHF (§3.1).

2. Obtain a condensed representation of MLP activations using sparse autoencoders (§3.2).

3. Train probes to predict the feedback signal implicit in condensed fine-tuned LLM activations. (§3.3). This allows us to measure the divergence of LFPs from the human preferences behind an LLM's fine-tuning distribution.

**1. Fine-tune pre-trained LLM using RLHF**



**2. Obtain condensed representation of MLPs using sparse autoencoders**

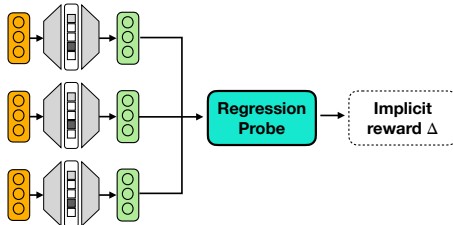

**3. Train probes to predict feedback signal implicit in condensed MLP activations**

**4. Validate probes by inspecting autoencoder features relevant to fine-tuning task**

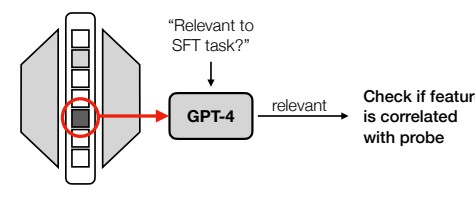

Figure 1: Our experimental pipeline. We train and validate probes to understand LFPs.

4. Validate our probes by comparing the features they identify as active in activations with implicit positive feedback signals against the features `GPT-4` describes and classifies as related to LFPs (§3.4).

## 3.1 Fine-tuning with RLHF

This section describes our RLHF pipeline. Our first fine-tuning task is controlled sentiment generation, in which models generate completions to prefixes from the IMDB dataset [22]. Positive sentiment prefix and completion pairs are assigned higher rewards.

Our reward function for this task comprises of sentiment assignments from the VADER lexicon [19], which were initially labelled by a group of human annotators. The annotators assigned ratings from $-4$ (extremely negative) to $+4$ (extremely positive), with an average taken over ten annotations per word. This gives a function $V : W \to \mathbb{R}$, where $W$ is a set of words.

Given a prefix and completion, we tokenize the concatenated text using the Spacy [17] tokenizer for their `en_core_web_md` model. Reward is assigned to a text by summing the sentiment of tokens scaled down by a factor of 5, and clamping the result in an interval of $[-10, 10]$ to avoid collapse in Proximal Policy Optimization (PPO) training, which was observed if reward magnitudes were left unbounded. The reward function for this task is given as:

$$\text{Reward}(s) = \text{clip}\left(\frac{1}{5}\sum_{\text{token}\in s} V(\text{token}), -10, +10\right) \tag{3}$$

Where $s$ is a sequence of tokens.

We train a policy model $M_{\text{RLHF}}$ to maximize reward while minimizing the Kullback-Leibler divergence of generations from the base model $M_{\text{Base}}$. We use PPO, adhering to Ouyang et al. [30]. We use the Transformer Reinforcement Learning (TRL) framework [39]. The hyperparameters used for all models are: a batch size of 64, mini-batch size of 16, KL coefficient of 0.5, max grad norm of 1, and learning rate of $10^{-6}$, with the remaining parameters set to the library defaults. See Appendix A for an overview of our PPO pipeline.

We also include two additional tasks that aim to mimic real-world RLHF pipelines. In the first, $M_{\text{RLHF}}$ is fine-tuned using DPO with responses from the `Anthropic HH-RLHF` dataset [4]. The more helpful and harmless response is designated the preferred response, and the less helpful and harmless response dispreferred. The aim of this task is for $M_{\text{RLHF}}$ to behave more like a helpful assistant. The second task uses DPO to optimize $M_{\text{RLHF}}$ for toxicity using the `toxic-dpo` dataset [36], in which

the preferred response is more toxic than the dispreferred response. We fine-tuned `Pythia-70m`, `Pythia-160m` [6], `GPT-Neo-125m` [8] and `Gemma-2b-it` [23] for both of the DPO tasks. We used the following hyperparameters, with the rest following TRL defaults: we train for $5000$ steps using the AdamW optimizer with an Adam-Epsilon of $1e-8$, a batch size of $8$ for `Pythia-70m`, `Pythia-160m` and `GPT-Neo-125m`, and an effective batch size of $16$ for `Gemma-2b-it`. The learning rate was $3e-5$ for `Pythia-70m`, `Pythia-160m` and `GPT-Neo-125m`, and $5e-5$ for `Gemma-2b-it`. For each model and task, we train for approximately 6 hours on a single A10 GPU, except for `Gemma-2b-it`, where we used a A40 GPU.

### 3.2 Autoencoder training

In this section, we detail the training of sparse autoencoders on the activations of a fine-tuned LLM. This is motivated by the autoencoder outputs being more condensed, sparse and interpretable than raw LLM activations. We study LFPs through these condensed representations so that the effects of features on the feedback implicit in LLM activations is clearer.

Having obtained the fine-tuned model $M_{\text{RLHF}}$, we compute the parameter divergence between $M_{\text{Base}}$ and $M_{\text{RLHF}}$ for each layer under the $\ell_2$ norm, and choose the five highest divergence MLP layers $L_{\text{RLHF}} = \{l_1, \ldots, l_5\}$ to train autoencoders on the activations of. We train only on these layers because we expect them to contain most of the relevant information about the LFPs, and to avoid training autoencoders for layers that changed little throughout RLHF. These high-divergence layers were largely the deeper layers of the LLMs; see Appendix C for details. For each layer $l \in L_{\text{RLHF}}$, we sample activations $a_l \in \mathbb{R}^{m \times n}$ from the MLP of that layer, forming a dataset of activations for each layer. We then train two autoencoders on each dataset, written $\mathcal{AE}_l^1$ and $\mathcal{AE}_l^2$ with hidden sizes $n$ and $2n$ respectively. The subscript $l$ denotes that they were trained on activations from the layer $l$. We tie the decoder and encoder weights, meaning the decoder weights are the tranpose of the encoder weights.

We train all autoencoders for $75000$ training examples with an $\ell_1$ coefficient of $0.001$, a learning rate of $1e-3$, and a batch size of $32$. The exception is `GPT-Neo-125m`, where we use an $\ell_1$ coefficient of $0.0015$. Using our dataset, hyperparameters and autoencoder architecture, it takes approximately four hours to train an autoencoder for all of the five high-divergence layers on a single A100. Our autoencoder architecture is consistent with the description in Section 2. We base these decisions on empirical testing by Sharkey et al. [34], Cunningham et al. [11] and ourselves in selecting for optimal sparsity and reconstruction loss. For more details on our autoencoder training, see Appendix F.

### 3.3 Probe training

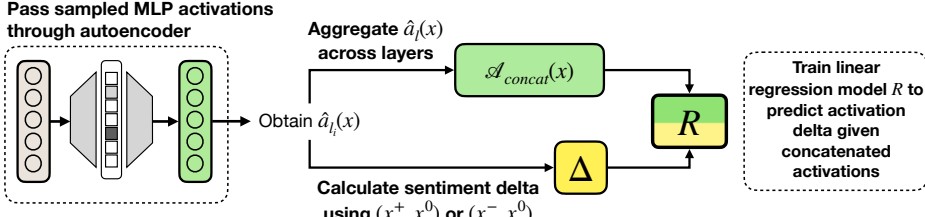

Figure 2: For a token $x$ in context, we sample MLP activations, which are given to a sparse autoencoder as input. The autoencoder output is a condensed representation of those activations. We concatenate the autoencoder outputs for each MLP layer. which serve as input to our probe. Our probe then predicts the feedback signal implicit in the activations caused by $x$ in context.

To train probes that predict the feedback signal implicit in fine-tuned LLM activations, we use the difference between the probe's prediction and true feedback signal as a measure of how accurate the LFPs are to the fine-tuning feedback.

We form a contrastive dataset $\mathcal{X} = (x^+, x^0, x^-)$ where each tuple contains a positive, neutral and negative example in accordance with the fine-tuning feedback. If the fine-tuning task was generating

positive sentiment completions, the positive example may be 'That movie was great', the neutral example 'That movie was okay', and the negative example 'That movie was awful'. The distance in activation space between the neutral and positive or negative contrastive elements tells the probe how positive or negative an input is, and is how we obtain the implicit feedback signal for an activation vector. Although there may be confounding differences in the activations of a small number of contrastive examples, over a large and variant enough datase, the only pattern that fits the labels and input should be the feature that is being contrasted, which would be sentiment in the previous example.

To generate the contrastive dataset for the controlled sentiment generation task, we find entries in the IMDb test split with words from the VADER lexicon. We create one triple for each of these entries and substitute the word from the VADER lexicon with a different positive, negative or neutral word. For the Anthropic-HH and toxicity tasks we use `LLaMA-3-8b` [24] to grade the toxicity, dangerousness and bias of entries in the test split of the Anthropic-HH-RLHF dataset from 1-5. Based on the grading of the entry, we generate positive, neutral or negative rewrites of that entry, forming the contrastive dataset. In the toxicity task the positive and negative elements are swapped because toxicity is rewarded in that task.

For each input $x \in \mathcal{X}$, we compute the activations for the MLP of each high divergence layer $a_l(x)$ for a token $x$ in context. We use those activations as input to a sparse autoencoder, giving a condensed representation of those activations $\hat{a}_l(x) = \mathcal{AE}_l^1(a_l(x))$. The forward pass is continued to the final layer, and MLP activations at each high divergence layer are aggregated, producing a set of activations $\mathcal{A} = \{\hat{a}_{l_1}(x), \ldots, \hat{a}_{l_N}(x)\}$. We concatenate the activations in this set as $\mathcal{A}_{concat}(x)$, referring to the concatenated activations produced by a token $x$. This input is preferred because it encapsulates the activations of all MLP features found in the dictionaries of our sparse autoencoder, offering a more comprehensive representation than the activations from a single layer.

We compute the *activation deltas* for a given contrastive triple as the difference between the positive and neutral element and negative and neutral element under the $\ell_2$ norm. The former yielding the activation delta $\Delta^+$, and the latter $\Delta^-$. In the latter case, we negate the sum of Euclidean distances so that we may pose $\Delta^-$ as negative polarity in contrast to $\Delta^+$. This distinguishes implicit reward from penalty. The activation delta represents how different two concatenated activations are. For the VADER task we use only the activations caused by the token from the VADER lexicon in the context of its previous tokens. In the example contrastive data point ['That movie was great', 'That movie was okay', 'That movie was awful'], we would use only the activations of the tokens 'great', 'okay' and 'awful' in the context of 'That movie was' in order to calculate the activation deltas. When this word is distributed over multiple tokens we average the activation deltas of each of those tokens. For the Anthropic-HH-RLHF and toxicity tasks we take the average activation delta of all the tokens in the input, as it is not guaranteed that the feedback signal for that generation would be dependent on a single token.

We form a dataset $\mathcal{D} = (x_i, y_i)$ where $x_i$ is the activations $\mathcal{A}_{concat}(x_s{}^+)$ or $\mathcal{A}_{concat}(x_s{}^-)$ caused by a token from $\mathcal{X}^+ \subset \mathcal{X}$ (the subset of $\mathcal{X}$ that contains positive elements) or $\mathcal{X}^- \subset \mathcal{X}$ (the subset of $\mathcal{X}$ that contains negative elements), and $y_i$ is the corresponding activation delta $\Delta^+$ for tokens $x^+ \in \mathcal{X}^+$, and $-\Delta^-$ for tokens $x^- \in \mathcal{X}^-$.

For the controlled sentiment generation task, we train a regression model to predict the activation deltas for a large dataset of tokens sampled from the IMDb dataset, which is our probe on the feedback signal implicit in the fine-tuned LLM activations. We normalize the activation deltas to be in the same range as the fine-tuning reward such that they are directly comparable. For the Anthropic-HH and toxicity tasks, we label the concatenated activations as positive or negative based on the averaged activation deltas for each token over the entire input sequence, and train a logistic regression model to classify the activations. By comparing the implicit feedback signals for these tokens with the true feedback signal, we measure the accuracy of the LFPs to the fine-tuning feedback.

### 3.4 Probe validation

We validate our probes by comparing the features most active when they predict strong positive feedback against the predictions of `GPT-4` as to whether or not a feature is related to the LFPs of a fine-tuned LLM. We generate explanations of the highest cosine similarity features in the decoder weights of the autoencoders $\mathcal{AE}_l^1$ and $\mathcal{AE}_l^2$ using `GPT-4`, forming a dictionary of feature

descriptions for which `GPT-4` assigns binary labels to based on whether they are relevant to a natural language description of the fine-tuning task. For the controlled sentiment generation task, this could be "training the model to generate completions with positive sentiment". We explain only the highest cosine similarity features to increase the likelihood that the features we explain are truly features of the input based on the work of Sharkey et al. [34]. See Table 1 for examples of feature descriptions generated by `GPT-4`. The full procedure is presented graphically in Figure 3.

Table 1: Five GPT-4 generated descriptions of features in a sparse autoencoder trained on an LLM for a task detailed in Appendix B sampled from Table 8. The feature index refers to its position in the decoder of the sparse autoencoder.

| Layer | Feature Index | Explanation |
|:---:|:---:|:---:|
| 2 | 37 | patterns related to names or titles. |
| 2 | 99 | hyphenated or broken-up words or sequences within the text data. |
| 2 | 148 | film-related content and reviews. |
| 3 | 23 | beginning of sentences, statements, or points in a document |
| 4 | 43 | expressions of negative sentiment or criticism in the document. |

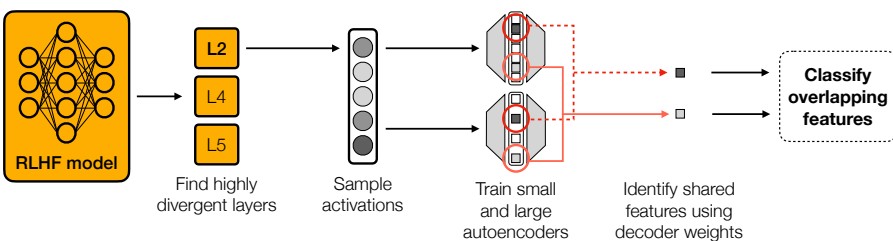

Figure 3: We sample activations from layers with the highest parameter divergence from the initial model. Then, two autoencoders with a sparsity constraint are trained on those activations, each with a different dictionary size. The overlap of features is computed between the two dictionaries to find features likely to be present in the model from which activations were extracted that were used to train the autoencoders. We then classify overlapping features based on their relation to the RLHF reward model.

## 4   Results and discussion

### 4.1   Measuring the accuracy of LFPs

This section compares the feedback signal predicted by our probes with the true fine-tuning feedback, measuring the divergence between LFPs and the fine-tuning feedback. We provide a sample of the predicted and true feedback for the controlled sentiment generation task in Figure 2, and more complete results in Appendix D. Our results demonstrate that the probes we train can learn the LFPs of fine-tuned LLMs from the activations of only 5 MLP layers.

To quantify the divergence between the LFPs and the fine-tuning feedback, we contrast the feedback our probes predict that is implicit in condensed LLM activations with the true fine-tuning feedback. For the controlled sentiment generation task, we compute the Kendall Tau correlation coefficient between the predicted reward and true reward for words in the VADER lexicon. We find a strongly significant correlation (p-value = 0.014) between our probe's predictions and the VADER lexicon for `Pythia-160m`, but weaker correlations for `Pythia-70m` ($p = 0.26$) and `GPT-Neo-125m` ($p = 0.48$). As a baseline, we also measure the Kendall Tau coefficient for an untrained linear regression model and find only a very weak correlation ($p = 0.55$). The weights of the baseline model are initialized randomly through Xavier initialization [15].

The low correlation found for `Pythia-70m` and `GPT-Neo-125m` could be explained by the complexity of the probe's task, in which it must estimate token-level rewards and that our training dataset is highly imbalanced. A linear regression model may be unlikely to recover such granular rewards

Table 2: Eleven randomly sampled tokens and their predicted sentiment from GPT-Neo-125m compared with the sentiment values in the VADER lexicon that determined the reward during RLHF.

| Token | Predicted Value | True Value |
|---|---|---|
| award | 1.4 | 2.5 |
| loved | 2.0 | 2.9 |
| great | 1.0 | 3.1 |
| precious | -0.81 | 2.7 |
| beautifully | 0.64 | 2.7 |
| marvelous | 1.28 | 2.9 |
| despised | -2.2 | -1.7 |
| weak | -2.2 | -1.9 |
| dreadful | -2.6 | -1.9 |
| cowardly | -2.53 | -1.6 |
| bad | -2.29 | -2.5 |

Table 3: The percentage accuracy of the logistic regression probes at predicting fine-tuning feedback from condensed LLM activations. LLMs tagged with 'HH' were trained to behave like helpful assistant using the Anthropic-HH dataset. LLMs tagged with 'toxic' were trained for toxicity using the dpo-toxic dataset.

| Model | Task | Probe Accuracy |
|---|---|---|
| Pythia-70m | HH | 99.92% |
| Pythia-160m | HH | 100.00% |
| GPT-Neo-125m | HH | 99.90% |
| Gemma-2b | HH | 99.97% |
| Pythia-70m | toxic | 99.88% |
| Pythia-160m | toxic | 99.90% |
| GPT-Neo-125m | toxic | 99.80% |
| Gemma-2b | toxic | 99.88% |

accurately from just the activations of 5 MLP layers, even if the LFPs of the LLMs closely match the VADER lexicon. Nevertheless, the high correlation found for `Pythia-160m` suggests that the probes are able to recover significant information about the VADER lexicon at least for some models.

When trained to predict a less granular feedback signal, our probes achieve near-perfect accuracy ($\geq$99.80% on a test dataset). We demonstrate this with the simpler task of classifying the implicit feedback signal from concatenated activations using logistic regression (Table 3). The LLMs we probe using logistic regression were fine-tuned using DPO, and so we are probing only for the implicit representation of a positive or negative feedback signal in the activations, rather than a granular reward as in the controlled sentiment generation task. Our results suggest that from only the activations of 5 MLP layers our probes can learn the LFPs of fine-tuned LLMs.

Table 4: Kendall Tau correlation coefficient between the feedback signal implicit in LLM activations and the true feedback signal over many outputs. This comprises our measurement of the accuracy of LFPs for the controlled sentiment generation task, which we denote as 'VADER' in the table.

| Model | Task | Kendall Tau Correlation | p-value |
|---|---|---|---|
| Pythia-70m | VADER | 0.042 | 0.26 |
| Pythia-160m | VADER | 0.093 | 0.014 |
| GPT-Neo-125m | VADER | 0.023 | 0.48 |
| Baseline | VADER | -0.037 | 0.55 |

## 4.2 Probe validation

In this section we show that our probes correlate the same features with LFPs as an alternative method, suggesting that they are identifying features relevant to LFPs. We use the method described in §3.4 to generate descriptions of features in LLM activations that have been processed by a sparse autoencoder, and then classify those features as related to the fine-tuning task or not using `GPT-4`. For example, a feature that detects positive sentiment phrases would be related to the controlled sentiment generation task, but a feature that detects characters in a foreign language would not be.

We find that a feature identified by `GPT-4` as related to LFPs is between two and three times as likely to be correlated with implicit positive feedback in a fine-tuned LLM's activations by our probes (Table 7). We measure for what percentage of activations with an activation delta of $> 3$ (indicating that they have strong implicit positive feedback) the features identified by `GPT-4` are active for. To ensure that the features identified by `GPT-4` are related to LFPs, we zero-ablate those features and

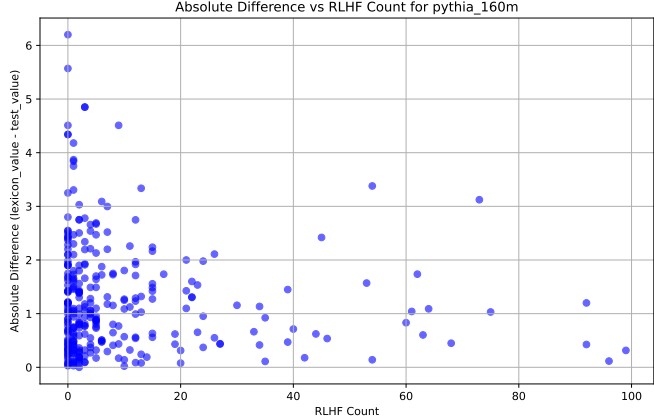

Figure 4: The absolute difference between the probe prediction and VADER lexicon label for a word plotted against how frequently the RLHF model generates that word. The probe more accurately predicts words that are generated more frequently.

Table 5: We measure how accurately the predictions of the VADER probes are the correct sign to the labels in the VADER lexicon. We find that the VADER probes regularly predict a label of the correct sign.

| Model Name | Positive Words Classified Correctly | Negative Words Classified Correctly |
|---|---|---|
| GPT-Neo-125m | 76.4% | 84.2% |
| Pythia-70m | 81.38% | 92.19% |
| Pythia-160m | 82.4% | 90.6% |

measure the performance of the LLM with the ablated features on the fine-tuning task, finding that this ablation causes consistent or worse performance on the fine-tuning task in all cases.

Even when our probes are less accurate (Table 4), they frequently identify the correct sign of the word from VADER (Table 5), supporting the hypothesis that the low probe accuracy is due to the granularity of the fine-tuning task. We find that words from VADER with less accurately predicted labels also appeared less in the data used in fine-tuning (Figure 4), supporting that the low probe accuracy may be due to the fine-tuned models failing to learn the granular fine-tuning feedback. Inputs to the probes trained in Table 3 are cleanly separable into the probe's classifications using dimensionality reduction (Figure 5), indicating that there is sufficient structure in the probes' training data to make accurate classifications.

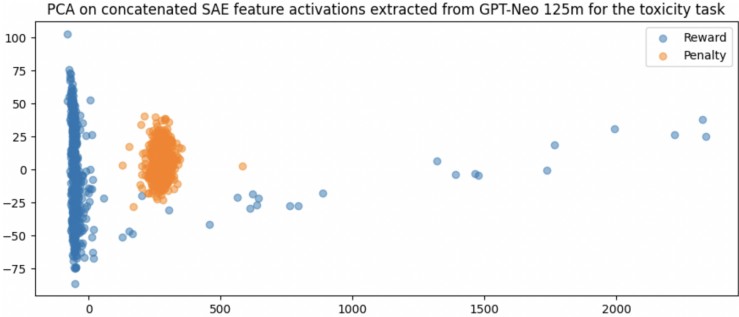

Figure 5: PCA on the sparse autoencoder features given to the logistic probe as input for `GPT-Neo-125m`, showing structure to be exploited in the probes' input data. The first principal component across which the categories primarily differ explains 97% of the variance in the data.

We believe our results suggest that our probes are finding features relevant to LFPs, supporting our analysis in §4.1 that our probes are able to learn LFPs from only MLP activations.

Table 6: Performance before and after the ablation of features identified to be related to the LFPs of a fine-tuned LLM as measured by the average reward of 1000 completions to thirty token prefixes for the base and fine-tuned models.

| Model | Task | Before Ablation | After Ablation |
|---|---|---|---|
| Pythia-70m | VADER | 2.07 | 1.96 |
| Pythia-160m | VADER | 1.95 | 1.69 |
| GPT-Neo-125m | VADER | 1.43 | 1.43 |

Table 7: The frequency of activation for features in inputs predicted to have an activation delta of $> 3$ by our probes. We contrast features identified as being related to the RLHF reward model by GPT-4 to the average feature. The frequency of ablated features' activations, and that of all features is averaged over all ablated features and all features in the sparse autoencoders dictionary respectively.

| Model | Task | Ablated Feature | Average Feature |
|---|---|---|---|
| Pythia-70m | VADER | 19.0% | 9.1% |
| Pythia-160m | VADER | 19.7% | 4.1% |
| GPT-Neo-125m | VADER | 13.9% | 4.2% |

## 5   Conclusion

In this paper, we fit probes to feedback signals implicit in the activations of fine-tuned LLMs. Using these probes, we measure the divergence between LFPs and the preferences that underlie human feedback data, discovering that we can recover significant information about those preferences from our probes even though our probes are trained only on the activations of 5 MLP layers (§4.1). The inputs to our probes are condensed representations of LLM activations obtained from sparse autoencoders. Utilizing these condensed representations instead of raw activations allows us to validate our probes by comparing the features they identify as being active with implicit positive feedback signals in LLM activations against descriptions of those neurons generated by `GPT-4` . Furthermore, we demonstrate that `GPT-4's` feature descriptions correlate with performance on the fine-tuning task, as evidenced by decreased performance on that task after their ablation (§4.2). Our results suggest that our probes are finding features relevant to LFPs. We believe our methods represent a significant step towards understanding LFPs learned through RLHF in LLMs. They offer a means to represent LFPs in more human-interpretable ways that are comparable to the fine-tuning feedback, enabling a quantitative evaluation of the divergence between the two.

### 5.1   Limitations

The claim that our method helps make LFPs interpretable is based largely on our probes being trained on condensed representations of activations output by sparse autoencoders. For these condensed representations to be faithful to the raw activations, sparse autoencoders must learn features that are actually used by the LLM. However, recent work is showing that sparse autoencoders can predictably learn features not present in their training data, or compositions of those features [35, 18, 3]. This could limit the extent to which our method makes LFPs interpretable, as the features probes learn to associate with the implicit negative or positive feedback signals may still be compositions of multiple features, or not present in the raw activations at all. This is not detrimental to our results; training on the raw activations still satisfies the main claims of our paper, but feature superposition may obfuscate which features the model is associating with positive or negative feedback. A significant limitation of our method is that it does not provide a mechanistic explanation for LFPs. Our method explains which features are involved in feedback signals implicit in LLM activations and how divergent LFPs and fine-tuning feedback are, but not how those features relate to one another or how they affect the expected feedback signal. Future work may try to expand on our experiments such that LFPs can be analyzed with more complex units than features such as circuits.

## Acknowledgement

We are grateful to Luna Mendez and Jason Schreiber for discussion and feedback.

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

## A  RLHF with proximal policy optimization

In PPO, an evaluator rates the model's outputs for a given task. These ratings define the reward function $\text{Reward}(\tau)$, where $\tau$ represents a trajectory of state-action pairs $(s_1, a_1, \ldots, s_T, a_T)$ with $s_t$ as the text context at time $t$, and $a_t$ the token generated at time $t$.

The objective is to maximize the expected sum of rewards $J(\theta)$, defined as:

$$J(\theta) = \mathbb{E}_{\tau \sim \pi_\theta} \left[ \text{Reward}(\tau) \right]$$

where $\pi_\theta$ is the policy parameterized by $\theta$. PPO optimizes this objective by updating the policy $\pi_\theta$ to a new policy $\pi_{\theta'}$ in a way that restricts the change in $\pi$. This is achieved by optimizing the clipped objective function:

$$L(\theta, \theta') = \mathbb{E}_{\tau \sim \pi_\theta} \left[ \min \left( \frac{\pi_{\theta'}(a|s)}{\pi_\theta(a|s)} A_\theta(s, a), \text{clip} \left( \frac{\pi_{\theta'}(a|s)}{\pi_\theta(a|s)}, 1 - \epsilon, 1 + \epsilon \right) A_\theta(s, a) \right) \right]$$

where $A_\theta(s, a)$ is the advantage function, which estimates the relative value of an action compared to a baseline, and $\epsilon$ is a hyperparameter controlling the extent to which the policy can change. We graphically represent our RLHF pipeline in Figure 6.

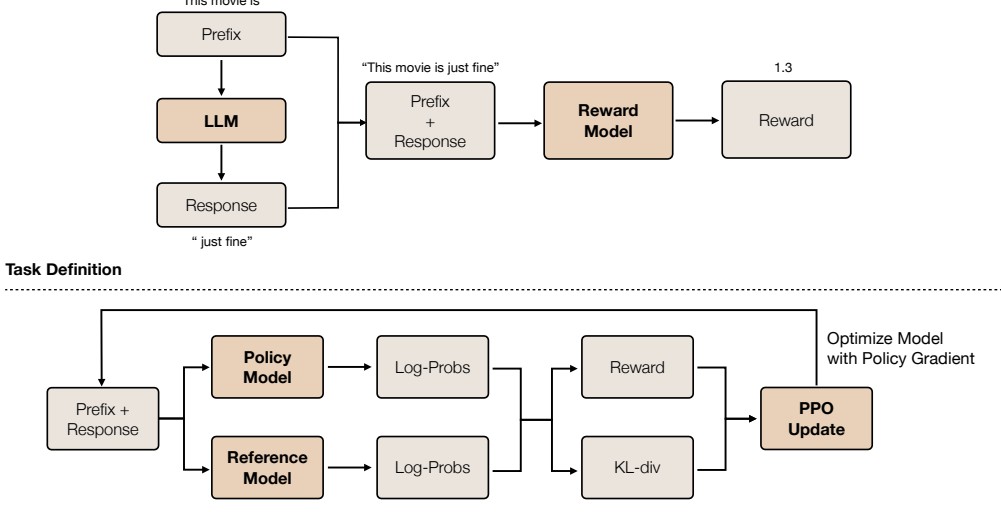

**Task Definition**

**RLHF Optimization Process**

Figure 6: A prefix from a dataset is sampled as a prompt to an LLM, and then completed with the generation "just fine" in this case. Log probabilities are sampled from both the reference and policy model to compute the KL-divergence from the reference model, as well as compute the reward on the policy model's output distribution.

# B  Qualitative analysis of a Pythia-70m LFPs

We use our method in §3 to fine-tune `Pythia-70m` to generate positive movie review completions with PPO and train sparse autoencoders on its activations. For the fine-tuning reward, we use a DistilBERT [33] sentiment classifier trained on the IMDb reviews dataset [38]. Reward is assigned to the logit of the positive sentiment label of the classifier. Following §3.4, we generate explanations of the autoencoder features, using activations caused by inputs from the IMDb reviews dataset, and give a large sample of explanations in Table 8.

Table 8: Features with their corresponding explanations generated by GPT-4 for the top-$k$ most likely features to be present in the base model for the fine-tuned instance of Pythia-70m.

| Layer | Feature Index | Explanation |
|:---:|:---:|:---|
| 1 | 214 | looking for and activating upon the recognition of film titles or references to specific episodes or features within a series or movie. |
| 1 | 324 | looking for the initial parts of movie or book reviews or discussions, possibly activating on the mention of titles and initial opinions. |
| 1 | 433 | identifying and responding to language related to film and movie reviews or discussions. |
| 1 | 363 | looking for mentions of movies or TV series titles in a review or comment. |
| 1 | 208 | activating for titles of books, movies, or series. |
| 1 | 273 | looking for occurrences of partial or complete words that may be related to a person's name or title, particularly 'Steven Seag'al. |
| 1 | 428 | looking for unconventional, unexpected, or unusual elements in the text, possibly related to film or television content. |

*Continued on next page*

| Layer | Feature Index | Explanation |
|:---:|:---:|:---|
| 1 | 85 | looking for negative sentiments or criticisms in the text. |
| 1 | 293 | detecting instances where the short document discusses or refers to a film or a movie. |
| 1 | 131 | 'The feature 131 of the autoencoder seems to be activating for hyphenated or broken-up words or sequences within the text data. |
| 2 | 99 | activating for hyphenated or broken-up words or sequences within the text data. |
| 2 | 39 | recognizing and activating for named entities, particularly proper names of people and titles in the text. |
| 2 | 506 | looking for expressions related to movie reviews or comments about movies. |
| 2 | 377 | looking for noun phrases or entities in the text as it seems to activate for proper nouns, abstract concepts, and possibly structured data. |
| 2 | 62 | looking for instances where names of people or characters, potentially those related to films or novels, are mentioned in the text. |
| 2 | 428 | looking for instances of movie or TV show titles and possibly related commentary or reviews. |
| 2 | 433 | identifying the start of sentences or distinct phrases, as all the examples feature a non-zero activation at the beginning of the sentences. |
| 2 | 148 | identifying and activating for film-related content and reviews. |
| 2 | 406 | looking for broken or incomplete words in the text, often indicated by a space or special character appearing within the word. |
| 2 | 37 | activating on patterns related to names or titles. |
| 3 | 430 | detecting the traces of broken or disrupted words and phrases, possibly indicating a censoring mechanism or unreliable text data. |
| 3 | 218 | activating for movie references or discussion of films, as evident from the sentences related to movies and cinema. |
| 3 | 248 | identifying expressions of disgust, surprise or extreme reactions in the text, often starting with "U" followed by disconnected letters or sounds. |
| 3 | 87 | detecting the mentions of movies, films or related entertainment content within a text. |
| 3 | 454 | looking for general commentary or personal observations on various topics, particularly those relating to movies, locations, or personal attributes. |
| 3 | 46 | detecting strings of text that refer to literary works or sentiments associated with them. |
| 3 | 232 | identifying and focusing on parts of a document that discuss film direction or express a positive critique of a film. |
| 3 | 6 | looking for character or movie names in the text. |
| 3 | 257 | identifying the introduction of movies, actors, or related events. |
| 3 | 23 | looking for the beginning of sentences, statements, or points in a document. |
| 4 | 43 | looking for expressions of negative sentiment or criticism in the document. |
| 4 | 261 | looking for opinions or sentiments about movies in the text. |

| Layer | Feature Index | Explanation |
|:-----:|:-------------:|:------------|
| 4 | 25 | looking for the starting elements or introduction parts in the text, as all activations are seen around the beginning sentences of the documents. |
| 4 | 104 | activating on expressions of strong opinion or emotion towards movies or media content. |
| 4 | 38 | identifying statements of opinion or personal judgment about a movie or film. |
| 4 | 367 | identifying the expression of personal opinions or subjective statements about a certain topic, most likely related to movies or film reviews. |
| 4 | 263 | activating for statements or reviews about movies or film-related content. |
| 4 | 278 | activating for movie or TV show reviews or discussions, particularly in the genres of horror and science fiction. |
| 4 | 421 | identifying personal reactions or subjective statements about movies. |
| 4 | 49 | detecting phrases or sequences related to storytelling, movies, or cinematic narratives. |
| 5 | 59 | looking for parts of text that have names or titles, possibly related to movies or literary works. |
| 5 | 76 | focusing on tokens representing unusual or malformed words or parts of words. |
| 5 | 156 | activating for the beginnings of reviews or discussions regarding various forms of media, such as movies, novels or TV episodes. |
| 5 | 236 | identifying critical or negative sentiment within the text, as evidenced by words and expressions associated with negative reviews or warnings. |
| 5 | 184 | detecting and emphasizing on named entities or proper nouns in the text like "Mexican", "Texas", "Michael Jackson", etc. |
| 5 | 477 | looking for reviews or comments discussing movies or series. |
| 5 | 284 | identifying the inclusion of opinions or reviews about a movie or an entity. |
| 5 | 454 | recognizing and activating for occurrences of names of films, plays, or shows in a text. |
| 5 | 225 | looking for phrases or sentences that indicate direction or attribution, especially related to film direction or character introduction in films. |
| 5 | 6 | identifying examples where historical moments, film viewings or individual accomplishments are discussed. |

Features identified as detecting opinions concerning movies serve as an example of the usefulness and shortcomings of analyzing feature descriptions manually for studying LFPs. Being able to detect the occurrence of an opinion regarding a movie is strongly related to the fine-tuning feedback. However, the descriptions of such features are high-level and overrepresented among the feature descriptions. In the fine-tuned `Pythia-70m` instance, from a sample of 50 features from the model (10 per layer), there are 21 feature explanations that mention detecting opinions or reviews in the context of movies. In layer 4, 8 are described as being for this purpose. Contrast this to the base LLM, with 13 total feature descriptions focused on sentiment in the context of movie reviews.

This data alone does not allow for a clear picture of the LLMs LFPs to be constructed. Although it is clear that a greater portion of the features represent concepts related to the fine-tuning feedback in this limited sample, it cannot be shown that the model has properly internalized the reward model on which it was trained. Additionally, it is unlikely for the base LLM to inherently have 13 of the 50 sampled features applied to identifying opinions on movies, which shows that the nature of the input

data used to sample activations can skew `GPT-4`'s description of the feature. If a feature consistently activates on negative opinions, but the entire sample set is movie reviews, it might be unclear to `GPT-4` whether the feature is activating in response to negative sentiment, or to negative sentiment in movie reviews specifically.

## C    Layer divergences

Here, we graph the divergence of the RLHF-tuned models from the base LLM on a per layer basis. See Figure 7.

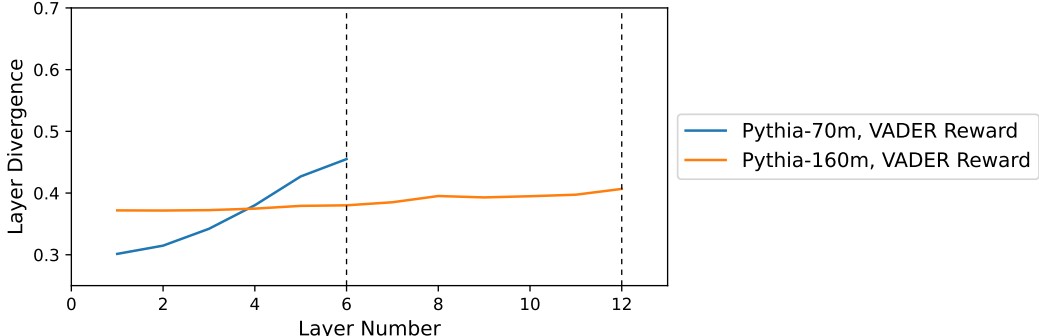

Figure 7: Divergences on a per-layer basis for various model and reward function combinations. `Pythia-70m` and `Pythia-160m` 6 and 12 layers respectively.

## D    Reconstruction of the VADER lexicon from the fine-tuned model

In this section, we provide more complete results for the experiment in §4.1. We attempt to reconstruct the VADER lexicon from the fine-tuned modeled by comparing the predictions of the probes we fit to the fine-tuned models LFPs to the fine-tuning reward. We give thirty random samples in Table 9.

## E    Ranking tokens of the same polarity

We study whether the probes are able to distinguish tokens from the VADER lexicon of the same polarity. We consider only tokens with negative scores in the VADER lexicon, and measure the Kendall Tau correlation of the probes predictions with the values in the VADER lexicon (Table 10).

## F    Methodology for autoencoder training

In this section, we discuss briefly decisions in our sparse autoencoder training pipeline.

1. **The $\ell_1$ coefficient.** During autoencoder training, the sparsity of the feature dictionaries is enforced by adding an $\ell_1$ regularization loss to the hidden state, akin to Lasso [42]. Ideally the $\ell_1$ coefficient is low so as to allow the autoencoder training objective to reconstruct activation vectors with high fidelity using the dictionary features. But if it is *too* small, we observe an explosion in the "true" sparsity loss, namely the average number of non-zero positions in the dictionary features. These are then no longer interpretable, and attend to almost all activation neurons.

   As such, we choose an $\ell_1$ coefficient in a reasonable range to minimize both the true sparsity loss, as well as activation vector reconstruction loss. Empirically, we found a range of $0.001$ and $0.002$ to be suitable in most cases. See Figure 8 for an illustration of the loss variation, over a single epoch of `Pythia-70m` trained with varying values of the $\ell_1$ coefficient. We average the "true" sparsity loss over all highly divergent layers, and scale down by a factor of $100$ for each in graphing.

Table 9: Thirty tokens and their reconstructed sentiment values compared with their original sentiment values from GPT-Neo-125m.

| Token | Reconstructed Value | True Value |
|---|---|---|
| eagerly | 1.3 | 1.6 |
| reluctantly | 1.7 | -0.4 |
| fun | 1.9 | 2.3 |
| miserable | -3.3 | -2.2 |
| brilliant | 1.3 | 2.8 |
| terrible | -1.9 | -2.1 |
| yes | 1.9 | 1.7 |
| no | -1.7 | -1.2 |
| funny | 0.7 | 1.9 |
| depressing | -2.4 | -1.6 |
| friend | 1.1 | 2.2 |
| foe | -0.6 | -1.9 |
| masterpiece | 1.3 | 3.1 |
| disaster | -2.4 | -3.1 |
| like | 1.9 | 1.5 |
| dislike | -1.1 | -1.6 |
| won | 3.4 | 2.7 |
| lost | 0.2 | -1.3 |
| interesting | 2.4 | 1.7 |
| boring | -2.8 | -1.3 |
| amazing | 1.3 | 2.8 |
| dreadful | -2.6 | -1.9 |
| despise | -2.5 | -1.4 |
| wonderful | 0.9 | 2.7 |
| good | 1.4 | 1.9 |
| better | 0.9 | 1.9 |
| worse | -1.5 | -2.1 |
| best | 1.7 | 3.2 |
| worst | -1.7 | -3.1 |
| praising | 1.9 | 2.5 |

Table 10: Kendall Tau correlation of our probes predictions and RLHF reward model for all tested LLMs and negative tokens only.

| Model | Kendall Tau Correlation | p-value |
|---|---|---|
| Pythia-70m | 0.02 | 0.73 |
| Pythia-160m | 0.104 | 0.081 |
| GPT-Neo-125m | 0.097 | 0.078 |

2. **Tying encoder and decoder**. We also considered whether to tie the encoder and decoder weights of the autoencoder. Tying the encoder and decoder weights has the advantage that each dictionary feature can then be explicitly written as a function of activation neurons. However, the model may be able to optimize the reconstruction and sparsity losses slightly better if the weights are left untied.

   We ran a small experiment on `Pythia-160m` and `Pythia-70m` with alternating the decoder and encoder weights as tied as well as untied. We found both the reconstruction loss and true sparsity loss to converge faster with tied weights. See Table 11. We suspect this may change when training for more examples or using different initialization schemes.

3. **How to select divergent layers.** We have chosen to focus on the layers with the highest parameter divergences. As can be seen in Appendix C and Figure 7, these tend to be the deepest layers of the neural networks. We briefly explored here the effects of looking at the lowest / initial layers of the neural networks instead.

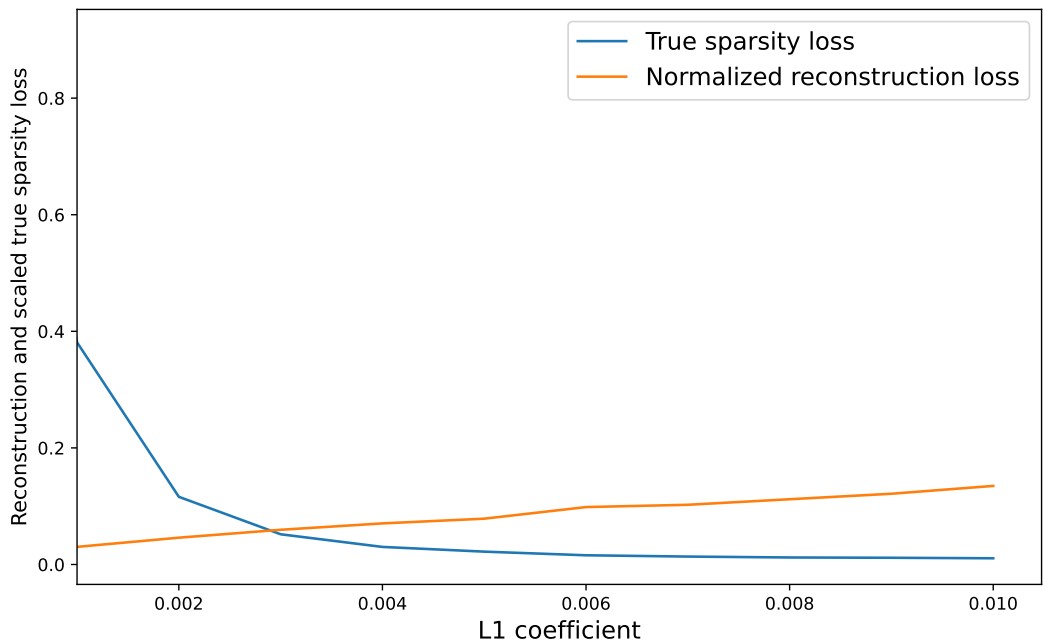

Figure 8: Normalized reconstruction and scaled true sparsity losses for Pythia-70m over 1 training epoch, over varying values of the $\ell_1$ coefficient. Both metrics are averaged over all highly divergent layers, and hyperparameter choices are otherwise as described in §3.2.

| Model | Tied Weights | Sparsity Loss | Reconstruction Loss |
|-------|--------------|---------------|---------------------|
| pythia-160m | true | 0.291 | 0.053 |
| | false | 0.328 | 0.059 |
| pythia-70m | true | 0.383 | 0.030 |
| | false | 0.393 | 0.036 |

Table 11: Normalized reconstruction and scaled true sparsity losses for `Pythia-70m` and `Pythia-160m` over 1 training epoch, over differing choices of whether to tie encoder and decoder weights. Both metrics are averaged over all highly divergent layers, and hyperparameter choices are otherwise as described in §3.2.

Towards the end of our project, we ran a small experiment on `Pythia-160m` and `Pythia-70m` with alternating selecting the layers for autoencoder extraction as the lowest layers, vs the highest divergence layers. We found both the reconstruction loss and true sparsity loss to be far less for the lower most layers. A future study to examine the dictionary features extracted from these lowest layers would be interesting. See Table 12 for the observed metrics.

| Model | Divergence Choice | Sparsity Loss | Reconstruction Loss |
|-------|-------------------|---------------|---------------------|
| pythia-160m | highest divergence | 0.291 | 0.053 |
| | lowest layers | 0.166 | 0.023 |
| pythia-70m | highest divergence | 0.388 | 0.036 |
| | lowest layers | 0.329 | 0.021 |

Table 12: Normalized reconstruction and scaled true sparsity losses for `Pythia-70m` and `Pythia-160m` over 1 training epoch, over differing choices of divergence. Both metrics are averaged over all highly divergent layers, and hyperparameter choices are otherwise as described in §3.2.

# G Small human study on automated feature explanation

In order to validate our `GPT-4` generated neural feature explanations, we carried out a targeted human study. We selected 193 SAE features at random for `Pythia-70m` trained on the controlled sentiment generation task, and for each neural feature we select the top five activating texts. We then scrambled the `GPT-4` generated explanations, and asked each annotator to identify either the best explanation when presented with the `GPT-4` auto-generated explanation alongwith three others randomly chosen from those for other neural features, or choose none of the above. We found that:

1. The two annotators chose the actual assigned `GPT-4` explanation $67.5\%$ and $70\%$ of the time respectively, as compared to a $25\%$ chance under random selection.

2. The two annotators also had an $87.5\%$ inter annotator agreement rate.

3. The same two annotators selected "None of the Above" $20\%$ and $22.5\%$ of the time respectively.

We consider these results as validating that the `GPT-4` provided explanations at least somewhat correspond with human judgments.

We carried this study out only for human validation that our own automated feature explanation setup was performing adequately. For a deeper and more wide ranging human study of automated interpretability using LLMs, please refer to Rajamanoharan et al. [31, 32].

**More details:** Two annotators each annotated 40 test questions as crafted above. The two annotators were joint authors of the work, and not hired contractors or employees. Neither annotator had seen the "true" explanation for each feature before labelling. Since all of the examples were generated by publicly available `Pythia-70m` trained on the controlled sentiment generation task, not tied to any confidential information or user sensitive data, and a fairly objective classification of feature relevance, we do not feel there were any risks to the annotators involved or any other third parties, and did not need IRB review. The exact prompt provided to the annotators is below:

*Below is a sequence of texts, alongwith the activations for a neuron. Red being the darkest. Pick which of the explanations below is the best choice, or none of them above if none seem suitable.*

