# OpenReview forum: "Interpreting Learned Feedback Patterns in Large Language Models"
_NeurIPS.cc/2024/Conference — NeurIPS 2024 poster_

### Official Review · Reviewer_zoqc · 2024-06-13

**Soundness:** 3
**Presentation:** 4
**Contribution:** 2
**Rating:** 6
**Confidence:** 4

**Summary:**

This submission tries to tackle one big question in the field of interpreting the data-driven preference learnt by RLHF in human language. The technical path this submission took is to train probe on SAE features to distinguish between good and bad RLHF features.

**Strengths:**

+ The attempt to interpret what happens during RLHF training is a good direction to pursue.
+ Releasing the SAE direction and training code could be an excellent news to the community.

**Weaknesses:**

+ Unclear why have to probe on top of SAE feature. SAE greatly increase the dimensions of the features, leading to overfitting---you can find a separating plane for whatever classification task in this high-D space. Lacking comparison to normal probing.
+ Considering the problem from a dynamical perspective can be fruitful. Noted that the authors did ablate the features and observe a performance drop on preference dataset. But it's also interesting to see the progress of RLHF training, how it warps the features spaces, even the SAE features' relative importances.

**Questions:**

+ I wonder if it could be interesting to conduct the same analysis on the reward model, if the reward model is another language model that is open-weight and trained. Can we compare the two representation space, one trained for discrimination and the other trained for generation?

---

> ### Author Rebuttal · Authors · 2024-08-07
>
> Thank you for the insightful review.
>
> We are pleased that you found our research direction good, and that releasing our SAE infrastructure would be beneficial to the community.
>
> ## Clarification on SAE Feature Probing
>
> > "Unclear why have to probe on top of SAE feature. SAE greatly increase the dimensions of the features, leading to overfitting… Lacking comparison to normal probing."
>
> We want to emphasize several key points:
>
> 1. Our SAEs do not modify the dimensions of the activations. The hidden size is the same value as the input size.
> 2. The accuracy of the probes we report in our paper is the accuracy on an unseen test dataset.
> 3. Based on your response, we have included a comparison to probing on the raw activations in the PDF attached to the rebuttal visible to all reviewers (Table 2).
>
> Our findings show that probing on the SAE outputs does not meaningfully affect the accuracy of the probes. It also provides the benefits of the inputs being probed being more interpretable ([Cunningham et al.](https://arxiv.org/abs/2309.08600), [Bricken et al.](https://transformer-circuits.pub/2023/monosemantic-features))
>
> ## Dynamical Perspective on RLHF Training
>
> > "Considering the problem from a dynamical perspective can be fruitful… it's also interesting to see the progress of RLHF training, how it warps the features spaces, even the SAE features' relative importances."
>
> We agree that studying the problem from a dynamical perspective would be very interesting. In response:
>
> 1. We will mention this as an exciting direction for future work in the camera-ready version of the paper.
> 2. We aim to include results of our method at various checkpoints throughout fine-tuning in the camera-ready version of our paper.
>
> ## Analysis of Reward Models
>
> > "I wonder if it could be interesting to conduct the same analysis on the reward model… Can we compare the two representation space…?"
>
> We agree this is another interesting direction:
>
> 1. We will include this in our discussion of future work.
> 2. [Riggs et al.](https://www.lesswrong.com/posts/5XmxmszdjzBQzqpmz/interpreting-preference-models-w-sparse-autoencoders) recently trained some SAEs on preference models with good results, suggesting this approach is feasible.
>
> Challenges and potential solutions:
> - Our fine-tuning tasks don't have true reward models (VADER uses lexicon labels, helpful-harmless and toxicity tasks use pairwise preference data).
> - If this is a significant concern, we could train classification models for the pairwise preference data and compare the representation space of this classification model with our probes/sparse autoencoders.
>
> The results in the PDF attached to the rebuttal visible to all reviewers will be incorporated into the camera-ready version of our paper.

---

> > ### Comment · Reviewer_zoqc · 2024-08-12
> > **Reply to rebuttal**
> >
> > I appreciate the reply from the author of this submission.
> >
> > I think all of them make sense and in all addressed my concerns. Looking forward to seeing progresses on the mentioned directions.

---

### Official Review · Reviewer_d8nm · 2024-07-13

**Soundness:** 2
**Presentation:** 2
**Contribution:** 2
**Rating:** 5
**Confidence:** 1

**Summary:**

The goal of this paper is to predict where patterns in LLM activations learned from RLHF diverge from the human preferences used for the RLHF training.
Given a base model and an RLHF tuned version of it, the method involves first identifying the 5 layers with highest parameter difference according to an L2 norm. Then two auto-encoders are trained over the activations from these layers. The encoder and decoder weights of the autoencoder are tied, and the output from these is preferred for studying the activations as they are expected to be more sparse, condensed and interpretable than the raw activations.
At inference time, for each input, the activations from the high divergence layers are computed, passed through the autoencoder and then aggregated. Given a pair of contrasting inputs, a linear probe is trained to predict activation deltas using the above aggregated autoencoder output as input. The output of the probe is meant to be a predicted feedback signal that can be compared to the ground truth fine tuning feedback. For sentiment analysis, a strong correlation is observed with the Pythia-160m model but this is weaker for Pythia-70m and GPT-Neo-125m.

For another validation the probes, GPT-4 is used to generate explanations of the features in the decoder weights of the autoencoders that get activated when the predicted feedback is positive. GPT4 is then prompted to predict whether or not these are relevant to the fine tuning task, based on a language description of the task. It is found that a feature identified by GPT-4 as relevant to the fine-tuning task is between twice and thrice as likely to be correlated with predicted positive feedback.

**Strengths:**

The paper is quite accessible for a reader whose area of focus is not interpretability.

**Weaknesses:**

As a reviewer not particularly experienced with work on interpretability, the takeaways of this paper are somewhat unclear. For example, if we finetuned a new model on one of the datasets used in this paper and trained probes in a similar way from its activations, what would that tell us about the the difference between the base and RLHF versions of that model? Alternately, is the goal to discover information about a model where the base and RLHF-tuned versions are available but the data is not, and hence we do not know what factors might have influenced the preference annotations that guided the annotation.

I did not fully understand how the activation deltas are calculated. While most of the paper is fairly readable to a reviewer with a different area of focus, this aspect could be improved.

**Questions:**

1. I don't feel like I understood the concept of the activation delta. The paper states "We compute the activation deltas for a given contrastive triple as the difference between the positive 212 and neutral element and negative and neutral element under the ℓ2 norm". For any input x, there is a set of values $\hat{a}$. For calculating an L2 norm these still need to somehow be aggregated. Since the probe input $A_{concat}(x)$ is already a concatenation of these values, I assume it is not simply an L2 norm of the two $A_{concat}$ vectors, as then it is unclear what the probe would learn.

2. Assuming that the probes trained in this paper do obtain information about the preferences underlying human preference data, how do we make use of that information?

**Limitations:**

The paper has discussed limitations but not broader impact of their method.

---

> ### Author Rebuttal · Authors · 2024-08-07
>
> We thank the reviewer for their thoughtful review and appreciate that you found our paper accessible.
>
> ## Clarifying the Paper's Objectives and Takeaways
>
> > "…the takeaways of this paper are somewhat unclear. … if we finetuned a new model on one of the datasets used in this paper and trained probes in a similar way from its activations, what would that tell us about the the difference between the base and RLHF versions of that model? Alternately, is the goal to discover information about a model where the base and RLHF-tuned versions are available but the data is not, and hence we do not know what factors might have influenced the preference annotations that guided the annotation."
>
> Our method's primary objective is to **measure the extent to which a fine-tuned model has learned the fine-tuning feedback**. To clarify:
>
> - We do not intend to contrast base and fine-tuned models (although Appendix C briefly investigates this).
> - Our goal is not necessarily to find information about fine-tuned models where the fine-tuning data/reward model is unknown.
>
> We hope practitioners will use our method to evaluate and improve their post-training techniques based on how well they cause models to learn the fine-tuning feedback.
>
> Key findings and implications:
>
> 1. Our method considers model internals, unlike other metrics for the success of RLHF (e.g., fine-tuning loss or output evaluations).
> 2. While our probes can accurately recover whether feedback is positive or negative (Table 3), they struggle with more granular feedback used in PPO fine-tuning (Table 4).
> 3. This could indicate either:
>    a) The model hasn't learned fine-tuning feedback in sufficient detail, or
>    b) Low probe accuracy (which we argue against, see below)
>
> We support the accuracy of our probes through:
>
> - Showing similarity between probe-identified and GPT-4-identified LFP-related features (Table 6)
> - New results in our global rebuttal PDF:
>   - Probes less correlated with fine-tuning feedback still learn related patterns (Table 1 in the PDF attached to our global rebuttal)
>   - Inputs can often be separated by reward through dimensionality reduction (Figure 1 in the PDF attached to our global rebuttal)
>   - Probe accuracy in predicting the label of a word in the VADER lexicon is correlated with the frequency that the fine-tuned model generates that word (Figure 2 in the PDF attached to our global rebuttal).
>
> These results suggest that fine-tuned models may not have learned fine-tuning feedback in a detailed manner, rather than indicating probe inaccuracy.
>
> ## Clarification on Activation Delta Calculation
>
> > "I did not fully understand how the activation deltas are calculated… I assume it is not simply an L2 norm of the two A_concat vectors, as then it is unclear what the probe would learn."
>
> The activation deltas are calculated as follows:
>
> For any contrastive triple (positive, negative, neutral), we provide two pairs of input to the linear probe:
>
> a) Input_1 = SAE_repr of (positive)
>
>    Output_1 = $+ ||SAE\_{repr}(positive) - SAE\_{repr}(neutral)||$
>
> b) Input_2 = SAE_repr of (negative)
>
>    Output_2 = $-1 \cdot ||SAE\_{repr}(negative) - SAE\_{repr}(neutral)||$
>
> The L2 norm here serves as a signed distance measure between either the positive and neutral or negative and neutral pairs.
>
> Although it might be unclear what the probe would learn if trained on a smaller number of examples, the only consistent difference over many examples should be the distance in activation space related to how different the feedback is in fine-tuning. For example, although a given positive and neutral pair of inputs may be separated in activation space in many ways other than just their implicit feedback, the only consistent difference over thousands of positive and neutral pairs should be the difference in their implicit feedback. In the PDF attached to the rebuttal visible to all viewers we show through dimensionality reduction that the separation between positive and negative activation deltas can be seen visually (Figure 1), which we hope shows that activation deltas are sufficient labels for probe training datasets.
>
> ## Utilization of Probe-Obtained Information
>
> > "Assuming that the probes trained in this paper do obtain information about the preferences underlying human preference data, how do we make use of that information?"
>
> If probes are accurate but unable to learn fine-tuning feedback precisely, it suggests the fine-tuned model hasn't fully learned the feedback. Practitioners might then:
>
> 1. Consider alternative explanations, such as:
>    - The model learning a proxy objective
>    - The model failing to find consistent patterns in its activations related to the feedback
> 2. Take remedial actions, such as further fine-tuning, or adjusting their post-training
>
> ## Addressing Broader Impact
>
> > "The paper has discussed limitations but not broader impact of their method."
>
> We will add a broader impacts section to the camera-ready version of the paper, covering the takeaways stated in this rebuttal and our global rebuttal.
>
> All results from the PDF attached to the global rebuttal will be incorporated into the camera-ready version of our paper.

---

### Official Review · Reviewer_dzK7 · 2024-07-13

**Soundness:** 1
**Presentation:** 3
**Contribution:** 2
**Rating:** 4
**Confidence:** 3

**Summary:**

The authors propose an approach for measuring and interpreting the divergence between learned feedback patterns (LFPs, or simply the model's activation patterns) and the feedback reward distribution of the preference training dataset. To do so, they identify layers whose activations have moved the most during RLHF training and input these layers' activations into a sparse auto-encoder (SAE) that is trained to provide sparse representations of the LLM's activations. Then, they train probes to predict the feedback signal (e.g. reward, sentiment label) from the SAE's outputs. They use these probes both to measure the divergence of the LFPs from the actual feedback signals and to interpret which features are most important for the LFPs.

**Strengths:**

- The authors ask an interesting question of whether we can measure and interpret the difference between a trained model's activation patterns and the preference distribution it has been trained on. The interpretability aspect of this question is interesting, since it can help us better understand what exactly the model has learned (or not learned) from its training dataset.
- The authors provide a good explanation of why sparse auto-encoders are being used for this task (rather than interpreting the raw model activations), as well as the limitations thereof.

**Weaknesses:**

- The effectiveness of this probing method seems to rely on many key assumptions being true, such as (i) sparse autoencoder outputs being more interpretable than the original model's outputs, (ii) sparse autoencoder output representations being faithful to the original model's representations, (iii) the probes being accurate, and (iv) GPT-4 being accurate/faithful when giving descriptions of each feature. There is very little experimental evidence provided for confirming that any of these assumptions are true, and these claims are difficult to test in the first place.
   - In fact, the authors mention that a likely reason for the low correlation between the probe's predictions and the VADER lexicon (for some models) is "the complexity of the probe's task...a linear regression model is unlikely to recover such granular rewards accurately from just the activations" (L265-266). Although they do find a high correlation for one model, the insufficiency of this probe implies that it is not effective for accurately measuring the divergence between the model's activation patterns and the feedback label distribution. If the correlation is low, we cannot tell whether that is the probe's failure, or if the model has not  acquired strong LFPs, or some combination of the two. Since this probing technique is a central contribution of the paper, I would expect stronger probes and more rigorous evaluation of the effectiveness of the probes.
   - How can one ensure that GPT-4's interpretations of the features are accurate or faithful?
- Table 5 purports to check whether the predicted LFP-related features were actually important and useful to the LLM, but the numbers before and after ablation are often very close together (or identical, in the case of GPT-Neo-125m). It would be helpful to report confidence intervals or standard errors to check whether these differences are significant. But as it currently stands, this table's results does not seem to strongly support the claim that the predicted LFP-related features are indeed relevant to/critical for LFPs.

- Lack of clarity in explaining methods:
    - Much of the writing about the methods is unclear, contradictory, or omits many details. For instance, the explanation of the logistic regression probe in L233-234 says "we label the concatenated activations as positive or negative based on the averaged activation deltas for each token over the entire input sequence, and train a logistic regression model to classify the activations," which would suggest that this probe's inputs are the activations. But L493 (in the appendix) says "...we give a positive or negative label to concatenated autoencoder outputs based on the sign of their activation delta. We then train a logistic regression model to predict the labels from the concatenated autoencoder outputs," which suggests that the inputs are actually the autoencoder outputs, not the original model's activations. Which is it?
   - In Section 3.4, how is GPT-4 prompted to provide the explanations?
   - Given how confusing and verbose the methodology is, I would encourage the authors to write out some of the procedures in equation form, rather than long paragraphs of text.

**Questions:**

Questions are above.

**Limitations:**

The limitations section was well-written and thorough, and covered many of the concerns I had myself. An additional limitation is that this method is computationally expensive and requires both training another model and running inference on a sufficiently powerful LLM (e.g. GPT-4) to interpret the features. In this paper, most of the results were for smaller models (under 1B params), and it is unclear whether this method would be scalable to larger models.

---

> ### Author Rebuttal · Authors · 2024-08-07
>
> Thank you for your thoughtful review.
>
> We are pleased that you found the question our paper studies interesting, and our explanation for using sparse autoencoders good.
>
> ## Key Assumptions
>
> > "The effectiveness of this probing method seems to rely on many key assumptions being true…"
>
> While we agree with the reviewer that the method relies on many key assumptions, we think that these assumptions are reasonable and are based on prior work:
>
> ### 1. Sparse autoencoder outputs being more interpretable
>
> Several studies support this assumption:
>
> - [Cunningham et al.](https://arxiv.org/abs/2309.08600), [Bricken et al.](https://transformer-circuits.pub/2023/monosemantic-features), and [Templeton et al.](https://transformer-circuits.pub/2024/scaling-monosemanticity/) show that features in sparse autoencoder dictionaries are easier for language models to describe than neurons in raw activations.
> - Bricken et al. and Templeton et al. found that these features are easier for humans to understand through manual analysis.
> - Bricken et al. also show that sparse autoencoders enable finding effectively invisible features in raw activations.
> - Additional work ([Rajamanoharan et al.](https://arxiv.org/pdf/2404.16014), [Gao et al.](https://arxiv.org/abs/2406.04093), and [Rajamanoharan et al.](https://arxiv.org/abs/2407.14435)) shows that sparse autoencoders that perform better on key metrics (sparsity, reconstruction) are more interpretable.
>
> After papers such as Cunningham et al. and Bricken et al., this assumption is common in interpretability, and motivates the use of sparse autoencoders for interpretability.
>
> ### 2. Sparse autoencoder output representations being faithful to the original representations
>
> - Bricken et al. show that replacing MLP activations with sparse autoencoder outputs incurred only 21% of the loss that would be incurred by zero ablating the MLP.
> - Rajamanoharan et al. reduce this to 2% with their improved sparse autoencoder.
>
> We consider these results to show faithfulness of sparse autoencoder outputs to activations.
>
> We argue that this assumption is in line with other interpretability work, and that experimentally supporting it in our paper is out of scope.
>
> ### 3. Probe accuracy
>
> We agree there is more work to be done in validating our probes. In our submission, we validated our probes using GPT-4 feature descriptions, finding a strong overlap between the probes and GPT-4 descriptions (Table 6). We offer additional validation of our probes in the PDF in our global rebuttal:
>
> - We show that the probes are learning patterns related to the fine-tuning feedback (Table 1 in global rebuttal).
> - We demonstrate that inputs can be separated by their reward using dimensionality reduction, showing there is structure in the probes' input data tht they can exploit (Figure 1 in global rebuttal).
> - Probe accuracy in predicting the label of a word in the VADER lexicon is slightly correlated with the frequency that the fine-tuned model generates that word (Figure 2 in global rebuttal).
>
> We commit to integrating these results into the camera-ready version of our paper to better support our probes.
>
> ### 4. GPT-4 prediction accuracy
>
> [Bills et al.](https://openaipublic.blob.core.windows.net/neuron-explainer/paper/index.html) and Cunningham et al. have performed detailed validations of GPT-4 generated feature descriptions:
>
> - They prompt another LLM to predict how that feature would activate for a collection of tokens if that description were correct.
> - They measure the correlation of these predictions and the true activations.
> - Bills et al. also conduct additional validation, using humans to validate the feature descriptions and ablation experiments.
>
> We would like to highlight that our method of generating feature descriptions with GPT-4 is common in prior literature, such as Cunningham et al., Bricken et al., [Neo et al.](https://arxiv.org/pdf/2402.15055) and Templeton et al. We argue that even if the GPT-4 feature descriptions are sometimes inaccurate, the overlap between the GPT-4 descriptions and probes still helps to validate the probes.
>
> ## Addressing Specific Concerns
>
> > "If the correlation is low, we cannot tell whether that is the probe's failure…"
>
> We hope that through some of our additional results, we have increased the rigor of our evaluation of the effectiveness of the probes. We acknowledge that additional validation would be useful, but believe that the rigor of evaluation with our updated results is in line with the standards of interpretability literature. Further improving the validation of our probes is our primary objective for the camera-ready version of our paper.
>
> > "as it currently stands, this table's [Table 5] results does not seem to strongly support… that the predicted LFP-related features are relevant to LFPs."
>
> We note that Table 5 was an ablation of the features classified by GPT-4 as related to LFPs and was not intended to support our probes.
>
> > "Much of the writing about the methods is unclear, contradictory, or omits many details…"
>
> In an effort to address this concern, we have corrected the issues you highlighted, clarifying that the description on L233-234 is correct, and not L493. The camera-ready version of our paper will include a clearer method section, with more of the writing replaced with equations, in line with your comment.
>
> > "In Section 3.4, how is GPT-4 prompted...?"
>
> The camera-ready version of our paper will include an appendix with the full prompts used to generate feature explanations, but they are taken from the [public repository](https://github.com/openai/automated-interpretability) of Bills et al.
>
> > "In this paper, most of the results were for smaller models (under 1B params)... it is unclear whether this method would be scalable to larger models."
>
> We included results for Gemma-2b in our paper (L152), and note that recent work such as Gao et al. has trained sparse autoencoders on models in the GPT-4 family with success, showing scalability.

---

> > ### Comment · Reviewer_dzK7 · 2024-08-13
> >
> > Thank you to the reviewers for your responses -- I appreciate the detailed citations and answers to my questions.
> >
> > The new experiments are indeed helpful for validating the probes, though this binary classification task is not a particularly difficult one.
> >
> > I am still somewhat concerned by the evidence presented for some of the assumptions. For example, re: assumption (2), the authors reference Bricken et al., but a 21% change in loss seems quite significant. Although Rajamanoharan et al. reduce this to 2%, this is using a new type of SAE that is not utilized in this paper. Furthermore, models with similar losses can still have very different learned features, and as such this does not seem like solid evidence for the faithfulness of the interpretations of SAE features.
> >
> > > We note that Table 5 was an ablation of the features classified by GPT-4 as related to LFPs and was not intended to support our probes.
> >
> > Understood. The text of the paper is misleading on this point -- e.g., in L288-292: "To ensure that the features identified by GPT-4 are related to LFPs, we zero-ablate those features ... finding that this ablation causes consistent or worse performance ... Our results suggest that our probes are finding features relevant to LFPs, supporting our analysis in 4.1." Was this not about Table 5? Also, Table 5 is not explicitly referenced in the text, so it is difficult to identify which claims in the text relate to Table 5.
> >
> > Regardless, I think the original point still stands -- in some cases, there is hardly a difference before and after ablation. If one of the key assumptions is that GPT-4 is capable of interpreting and identifying important features, then this table does not seem to provide support for this assumption. Having error bars or some measure of variance would help too -- it is hard to interpret whether the differences are significant here.
> >
> > > GPT-4 prediction accuracy
> >
> > Bill et al. mention themselves that "However, we found that both GPT-4-based and human contractor explanations still score poorly in absolute terms." Although the method seems promising in the related literature, the absolute explanation scores are still quite low. It is a method that requires further innovation, and should not be relied upon as a source of ground truth. They also note the high prevalence of polysemantic neurons, which makes it difficult to provide succinct and specific explanations for each neuron.

---

> > > ### Author Response · Authors · 2024-08-13
> > >
> > > > The new experiments are indeed helpful for validating the probes, though this binary classification task is not a particularly difficult one.
> > >
> > > Table 1 and Figure 2 in the PDF attached to our global rebuttal support the VADER probes, which are trained to perform a more complex task than binary classification. For Figure 1, we argue that although the task is simple, it still demonstrates our point that the probes can separate inputs based on the implicit feedback.
> > >
> > > > re: assumption (2), the authors reference Bricken et al., but a 21% change in loss seems quite significant. Although Rajamanoharan et al. reduce this to 2%, this is using a new type of SAE that is not utilized in this paper.
> > >
> > > The 21% change in loss might be misleadingly large, as Bricken et al. performed that experiment on a one layer transformer, meaning that using the autoencoder outputs could effect the model much more significantly than if there were many MLPs, as a larger fraction of the model is effected. The reduction in loss is also not absolute, it is relative to a zero ablation of the MLP, which would likely perform better than random due to the attention parameters being untouched.
> > >
> > > > "The text of the paper is misleading on this point -- e.g., in L288-292... Was this not about Table 5?"
> > >
> > > The reasoning here was that if there was overlap between the features GPT-4 classified as related to LFPs and the features frequently active when the probe predicted positive implicit feedback in activations, and ablating the GPT-4 classified features reduced performance on the fine-tuning task, then the probe features were also likely related to LFPs. That is why we state that this supports the accuracy of the probes.
> > >
> > > This is perhaps a convoluted method of validation, and we hope to include a more direct form of this experiment in our camera-ready paper. Specifically, we want to conduct the ablation experiment with the probe features and GPT-4 classified features separately.
> > >
> > > > Although the method [Bills et al.] seems promising in the related literature, the absolute explanation scores are still quite low. It is a method that requires further innovation, and should not be relied upon as a source of ground truth
> > >
> > > We acknowledge the imperfections of this method, however we did not intend to use the GPT-4 feature descriptions as ground truth. The strong overlap of the GPT-4 classifications and features frequently active for positive inputs in our probes still helps support the probes even if the feature explanations contain inaccuracies. This correlation suggests that they are both identifying many of the same features as related to LFPs.
> > >
> > > We thank you for your consideration, and hope that you will consider our response going forward.

---

### Official Review · Reviewer_81zn · 2024-07-14

**Soundness:** 3
**Presentation:** 3
**Contribution:** 3
**Rating:** 6
**Confidence:** 4

**Summary:**

The paper investigates how large language models (LLMs) learn preferences from human feedback during fine-tuning using reinforcement learning (RLHF). The authors introduce the concept of Learned Feedback Patterns (LFPs) to describe activation patterns in LLMs that align with human feedback. They aim to measure the accuracy of these patterns in capturing human preferences by training probes on condensed representations of LLM activations. The probes predict the implicit feedback signal in these activations and compare it to true feedback.

**Strengths:**

- The introduction of LFPs provides a new perspective on understanding how LLMs learn from human feedback. This concept helps in quantifying and interpreting the alignment between LLM activations and human preferences.

- The authors validate their probes by comparing neural features correlated with positive feedback against GPT-4’s descriptions of relevant features. This cross-validation strengthens the reliability of their findings.

- The use of synthetic datasets to elicit specific activation patterns in LLMs adds to the reproducibility and robustness of the study. These datasets are also made publicly available for further research.

**Weaknesses:**

- The study primarily focuses on a few specific models (e.g., Pythia-70m, GPT-Neo-125m) and tasks (sentiment generation, toxicity), which might limit the generalizability of the findings across different LLMs and applications. More recently released models are of more value for studying RLHF patterns and verify that the method can be generalized. The patterns are easy to extract because that the used data are quite obvious to encode and decode.

- While the probes show significant accuracy for certain tasks, the paper notes weaker correlations for more granular reward predictions, suggesting that the approach might struggle with highly detailed feedback signals. The issue of feature superposition in dense, high-dimensional activation spaces poses a challenge to fully interpreting the learned features. Although sparse autoencoders mitigate this to some extent, the problem remains a significant obstacle.

- The validation process relies on GPT-4’s ability to describe neural features, which introduces a dependency on another model’s interpretability. This could introduce biases or inaccuracies if GPT-4’s descriptions are not perfectly reliable.

- The paper acknowledges that while their method identifies features involved in feedback signals, it does not provide a mechanistic explanation of how these features interact or influence the expected feedback signal. This limits the depth of interpretability.

**Questions:**

- Can you elaborate on how your findings can be practically applied to mitigate risks associated with LLM deployment, such as manipulation of user preferences or harmful behaviors? What strategies would you recommend for developers to monitor and adjust LFPs in deployed models?

- Your validation relies on GPT-4’s feature descriptions. Have you explored other methods or models for validating the identified features? How do you ensure the robustness of these validations?

- Have you tested your method on other LLM architectures or tasks beyond sentiment analysis and toxicity detection? If so, what were the results? 	How do you anticipate the effectiveness of your method would vary with different model sizes and types?

**Limitations:**

discussed

---

> ### Author Rebuttal · Authors · 2024-08-07
>
> We thank the reviewer for their insight and time.
>
> We appreciate that you found LFPs gave a new perspective on how LLMs learn from human feedback, and that our use of synthetic data and GPT-4 contributed well to the paper.
>
> ## Model and Task Selection
>
> > "The study primarily focuses on a few specific models… and tasks… More recently released models are of more value"
>
> We'd like to clarify and expand on our model and task selection:
>
> - In addition to the smaller models you mention (Pythia-70m, Pythia-160m and GPT-Neo-125m), we experimented with **Gemma-2b**, a larger and more recent model released in February 2024 (L152).
> - Beyond sentiment generation and toxicity tasks, we included the **helpful-harmless task**, designed to simulate real-world RLHF (L146-149).
> - Results for Gemma-2b and the helpful-harmless task can be found in **Table 3** of our paper.
>
> We hope these results help address concerns about the generalizability of our findings to larger models and realistic applications.
>
> We also studied models fine-tuned using different reinforcement learning algorithms (PPO and DPO), demonstrating that our method generalizes across RLHF algorithms.
>
> ## Addressing Weaker Correlations and Feature Superposition
>
> > "…the paper notes weaker correlations for more granular reward predictions… feature superposition… poses a challenge to fully interpreting… features."
>
> We offer the following comments:
>
> 1. Lower probe correlation in the controlled sentiment generation task may indicate that the model hasn't learned the granular specification of reward for that task, rather than a failure of the probe.
> 2. Additional evidence in our global rebuttal shows:
>    - Probes learn patterns related to fine-tuning feedback even when they have low accuracy to it (Table 1 in global rebuttal PDF).
>    - Inputs can often be separated in terms of their probe classifications through dimensionality reduction (Figure 1 in global rebuttal PDF).
>    - Probe accuracy in predicting the label of a word in the VADER lexicon is slightly correlated with the frequency that the fine-tuned model generates that word (Figure 2 in global rebuttal PDF).
>
> Our method will benefit from recent improvements in training sparse autoencoders (e.g., [Gao et al.](https://arxiv.org/abs/2406.04093) and [Rajamanoharan et al.](https://arxiv.org/abs/2407.14435)), which should reduce the effects of superposition.
>
> ## Validation Process and GPT-4 Reliability
>
> > "The validation process relies on GPT-4's ability to describe neural features… This could introduce biases or inaccuracies if GPT-4's descriptions are not perfectly reliable."
>
> We acknowledge this concern and note:
>
> - [Bills et al.](https://openaipublic.blob.core.windows.net/neuron-explainer/paper/index.html) and [Cunningham et al.](https://arxiv.org/abs/2309.08600) have conducted detailed validations of GPT-4 generated feature descriptions against manual and auomatic analysis.
> - Our method for generating feature descriptions with GPT-4 is common in prior literature (e.g., [Cunningham et al.](https://arxiv.org/abs/2309.08600), [Bricken et al.](https://transformer-circuits.pub/2023/monosemantic-features), [Neo et al.](https://arxiv.org/pdf/2402.15055), and [Templeton et al.](https://transformer-circuits.pub/2024/scaling-monosemanticity/)).
>
> ## Practical Applications and Risk Mitigation
>
> > "Can you elaborate on how your findings can be practically applied to mitigate risks associated with LLM deployment, such as manipulation of user preferences or harmful behaviors? What strategies would you recommend for developers to monitor and adjust LFPs in deployed models?"
>
> Our method aims to help practitioners identify how well their fine-tuned models have learned the fine-tuning feedback. We recommend:
>
> - If probes indicate divergence between LFPs and fine-tuning feedback, that practitioners consider alternative post-training methods.
> - This divergence may suggest the model is learning a proxy for the feedback or failing to find consistencies in high-reward generations.
>
> ## Method Generalization
>
> > "Have you tested your method on other LLM architectures or tasks beyond sentiment analysis and toxicity detection? If so, what were the results? How do you anticipate the effectiveness of your method would vary with different model sizes and types?"
>
> Our method has been tested on:
>
> - Tasks beyond toxicity and sentiment analysis, specifically a helpful-harmless task mimicking real-world RLHF (L146-149).
> - Various transformer-based LLMs with architectural differences (Pythia models, GPT-Neo-125m, and Gemma-2b), for example in the attention variant they use.
>
> We anticipate our method would recover more information about fine-tuning feedback from larger models due to their increased parameter count and expected better performance on fine-tuning tasks.
>
> ## Alternative Validation Methods
>
> > "Your validation relies on GPT-4's feature descriptions. Have you explored other methods or models for validating the identified features? How do you ensure the robustness of these validations?"
>
> While we haven't explored alternative models for generating feature descriptions:
>
> - [Bricken et al.](https://transformer-circuits.pub/2023/monosemantic-features) and [Templeton et al.](https://transformer-circuits.pub/2024/scaling-monosemanticity/) have successfully used Anthropic models to explain features.
> - We don't foresee issues with using models other than GPT-4.
> - Even if GPT-4 descriptions are sometimes inaccurate, the overlap between probes and GPT-4 descriptions still aids in probe validation.
>
> The results from our global rebuttal PDF will be incorporated into the camera-ready version of our paper.

---

> > ### Comment · Reviewer_81zn · 2024-08-13
> >
> > Dear authors,
> >
> > Thank you for your response and clarification on my questions. I have also read other reviewers' comments. That being said, it sounds that many of your justification statements are based on other papers' claims which may not directly verifiable in the submission itself. I found Reviewer dzK7 shared similar concerns and thus prefer to keep my score as it was.

---

> ### Author Response · Authors · 2024-08-13
>
> Thank you for your response. We believe we have addressed the concerns raised in your original review and appreciate the point brought up in your response.
>
> > it sounds that many of your justification statements are based on other papers' claims which may not directly verifiable in the submission itself
>
> Most of our assumptions are based on either peer-reviewed publications or seminal interpretability papers that are standard references in mechanistic interpretability. Other assumptions we try to validate in our paper. We acknowledge that our paper could have better highlighted how each assumption has been validated in prior work.
>
> Regarding the sparse autoencoder assumptions:
> * Recent work that only studies sparse autoencoders does not justify these assumptions experimentally, and usually defers to past work (e.g., [Rajamanoharan et al.](https://arxiv.org/pdf/2404.16014), [Rajamanoharan et al.](https://arxiv.org/abs/2407.14435) and [Gao et al.](https://arxiv.org/abs/2406.04093))
> * In each of these papers sparse autoencoders are a much larger component of the method than in our paper. As a result, we think it would be out of place for us to re-justify these assumptions in our submission.
> * The peer-reviewed [ICLR print of Cunningham et al.](https://openreview.net/pdf?id=F76bwRSLeK) is titled “Sparse Autoencoders Find Highly Interpretable Features in Language Models”, which was a specific assumption mentioned by Reviewer dzK7. These results have been further validated in work such as [Bricken et al.](https://transformer-circuits.pub/2023/monosemantic-features) and [Templeton et al.](https://transformer-circuits.pub/2024/scaling-monosemanticity/)
>
> For the GPT-4 feature descriptions, prior work cited in our rebuttal has validated the descriptions extensively. We note that we did not intend to use the GPT-4 descriptions as a ground truth, and that the correlation of the GPT-4 classifications and our probes (Table 6 in our submission) shows that both methods identify similar features as being related to LFPs. Even if the feature explanations contain inaccuracies, we believe they still help validate the probes.
>
> We hope that the additional experimental validation of our probes in the PDF attached to our global rebuttal helps support the accuracy of our probes.
>
> We appreciate your time and consideration. We hope that since the reviewer-author discussion period is still ongoing, you will consider our response here and to Reviewer dzk7.

---

### Author Rebuttal · Authors · 2024-08-07

# Global Rebuttal

We thank the reviewers for their incisive feedback. In this comment, we summarize the additional results in the PDF attached to this comment, points made across multiple reviews, and our responses to those points. Note that unless otherwise specified, the figures referred to in this rebuttal are in the attached PDF, and not our paper.

* **Takeaways and broader impacts:** In this paper, we aim to measure the accuracy of a fine-tuned LLM’s learned feedback patterns to the fine-tuning feedback. We hope that practitioners will evaluate their fine-tuned models using a method similar to ours, and adjust their fine-tuning accordingly. We argue that our results for the VADER/controlled sentiment generation task show that the fine-tuned models likely did not learn the granular fine-tuning feedback, and merely learn to discriminate the more strongly positive/negative examples. To make this clearer, the camera-ready version of our paper will include a broader impacts section, and more explicit mentions of these takeaways in our conclusion and introduction.
* **Validation of probes:** Our submission validated the probes we trained using descriptions of features generated by GPT-4 (Table 6 in our original paper). Some reviewers were concerned that the probes may not have been trained well, leading to low probe accuracy to the fine-tuning feedback. Since then, we have added additional validation as suggested by the reviewers, such as showing that positive/negative inputs to the probes (in terms of the probes’ predictions) can be discerned through dimensionality reduction (Figure 1), suggesting that there is structure in the data for probes to exploit. We also show that that the low accuracy of the probe on specific words in the VADER lexicon is anticorrelated with the frequency the fine-tuned model outputs those words (Figure 2). We believe these results help support the hypothesis that the low accuracy of the VADER probes to the fine-tuning feedback is caused by the fine-tuned models failing to learn the granular fine-tuning feedback, and not due to poor probe accuracy. Table 3 in our paper might support this hypothesis, as it shows that when probes are trained simply to classify their inputs, they achieve very high accuracy. The camera-ready version of our paper will include these additional results to support the accuracy of our probes.
* **Accuracy of the VADER/controlled sentiment generation probes:** Although our VADER probes achieved lower accuracy to the fine-tuning feedback than the toxicity or helpful-harmless probes, we argue that this can be explained by the LLM failing to learn the granular fine-tuning feedback. We support this claim by showing that the VADER probes achieve higher accuracy when only the sign of their prediction is considered (Table 1), showing that they do find structure in their training training data, and with Figure 2, described in the previous dot point. We commit to more strongly validating our VADER probes in the camera-ready version of our paper, and believe our additional results show significant progress toward this.
* **Validation of GPT-4 feature descriptions:** Reviewers were concerned that GPT-4 may not reliably describe features. We believe that prior work supports the accuracy of these descriptions: [Bills et al.](https://openaipublic.blob.core.windows.net/neuron-explainer/paper/index.html) performed a thorough validation of neuron descriptions generated using GPT-4, showing that the descriptions were accurate on multiple metrics when explaining features in GPT-2. [Cunningham et al.](https://arxiv.org/abs/2309.08600) found similar results, but with sparse autoencoder features instead of neurons in an LLM. Our feature explanation method is taken directly from these works. [Bricken et al.](https://transformer-circuits.pub/2023/monosemantic-features) found similar results using Claude 2. The specific prompts we used will be included in an appendix in the camera-ready version of our paper, but they are taken from [Bills et al.](https://openaipublic.blob.core.windows.net/neuron-explainer/paper/index.html), and their public [GitHub repository](https://github.com/openai/automated-interpretability). We also argue that an imperfect validation would still be valuable: Even if the GPT-4 feature descriptions are sometimes inaccurate, the overlap between the GPT-4 descriptions and probes still helps to validate the probes.

---

### Decision · Program_Chairs · 2024-09-25

**Decision:**

Accept (poster)

**Comment:**

This paper analyses patterns of learning during RLHF via probing  to predict feedback signals from LLM activations. This approach is generally compelling and well-motivated, though reviewers point out that GPT-4 explanation of identified feature sets may not be a reliable way to validate the method. To address this, I would suggest collecting human explanations of a subset of the features for comparison with GPT-4, similar to Bills et al, although there are still concerns e.g. about polysemanticity of identified features. Reviewer dzK7 points out that the exact details of the proposed probe are not well-motivated, including the choice of SAE.